# Microsatellite Markers: A Tool to Assess the Genetic Diversity of Yellow Mustard (*Sinapis alba* L.)

**DOI:** 10.3390/plants12234026

**Published:** 2023-11-29

**Authors:** Eva Jozová, Michael Rost, Andrea Rychlá, Dagmar Stehlíková, Baveesh Pudhuvai, Ondřej Hejna, Pavel Beran, Vladislav Čurn, Miroslav Klíma

**Affiliations:** 1Department of Genetics and Biotechnology, Faculty of Agriculture and Technology, University of South Bohemia in Ceske Budejovice, 370 05 Ceske Budejovice, Czech Republic; rost@fzt.jcu.cz (M.R.); stehld@fzt.jcu.cz (D.S.); bpodhuvai@fzt.jcu.cz (B.P.); hejna@fzt.jcu.cz (O.H.); beranp@fzt.jcu.cz (P.B.); curn@fzt.jcu.cz (V.Č.); 2OSEVA Development and Research, 746 01 Opava, Czech Republic; rychla@oseva.cz; 3Crop Research Institute, 161 06 Prague, Czech Republic; klima@vurv.cz

**Keywords:** molecular markers, SSR, breeding, genetic structure, induced mutagenesis

## Abstract

Microsatellite markers were used for the assessment of genetic diversity and genetic structure in a germplasm collection of yellow mustard, *Sinapis alba* L. The comprehensive collection of genetic resources represented 187 registered varieties, landraces, and breeding materials. Microsatellites generated 44 polymorphic alleles in 15 loci. Eleven of them were medium to highly polymorphic, and the high levels of observed heterozygosity (0.12–0.83) and Nei’s gene diversity index (0.11–0.68) indicated a high level of polymorphism. Based on PCoA and neighbor joining analyses, the genetic resources were divided into two groups. The range of genetic dissimilarity in the analysed collection was in the range of 0.00–1.00. The high level of dissimilarity between the accessions was documented by the high WAM value (33.82%). Bayesian clustering algorithms were performed in the STRUCTURE 2.3.4 software. The number of clusters was estimated at K = 2. The accessions were classified according to Q1/Q2 values. The low average values of the parameters Fst_1 (0.3482), Fst_2 (0.1916), and parameter alpha (0.0602) indicated substantial mating barriers between varieties and reproductive isolation due to the limited exchange of genetic resources between breeders. These results demonstrated the importance of extensive collections of genetic resources for the maintenance of genetic diversity and indicated considerable genetic differentiation among accessions.

## 1. Introduction

Yellow mustard (*Sinapis alba* L.) is a vital crop for Czech agriculture’s food and condiment sectors. It originated in the Middle East and Mediterranean regions. Still, this crop has been produced worldwide for millennia as a condiment crop. It is extensively planted as a prominent speciality crop [1], and the seeds and volatile oils are used in traditional Chinese medicine [2]. This crop is drought-tolerant and resistant to major diseases and insect pests [3,4]. Yellow mustard also has phytosanitary properties and can potentially reduce harmful nematodes in soils, and the soil disinfestation effects of biofumigant compounds released from yellow mustard green manure have been reported [5]. Complete data from the literature and original data on *S. alba* exposed to elements such as metals from the thallium, arsenic, and platinum groups were collected, ensuring crop application for phytoremediation in contaminated areas [6]. The Czech Republic is at the forefront of this crop production, due to ideal climatic conditions and a long heritage in cultivation. Together with Canada and Ukraine, the Czech Republic is one of the world’s largest producers and exporters of this crop. Yellow mustard is cultivated mainly for seeds and non-production purposes, such as an ameliorating catch crop for green manuring or as a significant component of biobelts. Mustard is also an important bee nectar and pollen-producing plant [7].

The range of genetic diversity, changes in genetic structure over the timespan of breeding in landraces, cultivated varieties, and genetic resources are still little understood, even though yellow mustard, like rapeseed, is a vital crop [4,8]. There is a widespread perception that the genetic variability is declining, due to breeders’ emphasis on production and product/seed quality, compared to other intensively cultivated crops [9]. Morphological traits, previously the only tool for selection and cultivar identification, are no longer sufficient to identify emerging varieties and define their phenotypic variance, due to the influences of many environmental factors. This is also reflected in the often problematic use of DUS (Distinctness, Uniformity and Stability) tests for the description and legal protection of varieties [10].

The unremitting reduction in the number of cultivated species and varieties and their widespread loss has led to genetic erosion, the partial or total loss of landrace varieties and a narrowing of genetic diversity. The effects of a lack of genetic diversity can be seen in the loss of important genes and genetic vulnerability, as well as in the reduction in genetic gain in quantitative traits, which is difficult to overcome [9,10,11]. The breeders then have then sought to expand existing genetic diversity, e.g., by using landrace varieties, by distant hybridisation with taxonomically more distant species, and by applying in vitro techniques and targeted mutagenesis [12]. In practical breeding, the use of classical mutagenesis is limited, and only a few new varieties have been bred based on this approach. Because of the difficult detection of mutations and the not-quite-successful use of induced mutagenesis in plant breeding, a molecular or reverse genetics approach has been adopted since the beginning of the millennium through the TILLING (targeting induced local lesions in genome) method [13]. A more modern mutant detection procedure involves amplifying genes of interest, sequencing amplicons, and identifying point mutations [14].

Plant genetic resources should be characterised by employing molecular markers to reveal genotypic variations and unambiguously identify crop varieties. Molecular methods have been essential components of most research on genetic diversity over the last few decades, and molecular markers recognise DNA polymorphism between varieties and aid in gaining a better understanding of the genetic hierarchy of the parent components [15]. Among the most used types of markers to assess genetic diversity or seed purity are, for example, RAPDs (Random Amplified Polymorphic DNA) [16], which are no longer widely used due to poor repeatability of results, AFLP (Amplified Fragment Length Polymorphism) [17], ISSR (Inter Simple Sequence Repeats) [18], SSR (Simple Sequence Repeats) [4,19,20], or SNP (Single Nucleotide Polymorphism) [21]. However, it is essential to recognise that different markers have distinct attributes/properties that reflect various components of genetic variation and, therefore, can produce different results [22]. Although yellow mustard is an important crop, the number of studies devoted to this species is quite limited. A detailed study of yellow mustard accessions, maintained at Plant Gene Resources of Canada, was performed using AFLP markers [17]. The same collection of genetic resources was recently analysed using a modern approach to genotyping by sequencing, and new sets of SNPs were discovered and evaluated [21]. More attention was focused on coloured mustard, *B. juncea*, *B. carinata*, and *B. nigra*. An SSR marker was used in *B. juncea* [23] and *B. nigra* [24]. In *B. juncea*, the influence of natural outcrossing and cross-hybridization for the development of analysed varieties was demonstrated, based on the analysis of the genetic structure. Unique SSR alleles were used for specific DNA fingerprinting for the identification and legal protection of *B. juncea* varieties. Genome-specific SSR markers in *B. nigra* were used for assessment of the correct taxonomic classification of accessions. Based on multidimensional scaling analyses, genotypes were associated with country or region of origin. In *B. carinata* [25], the genetic diversity and population structure of genotypes of different origins was analysed using high-throughput SNP markers. The analysed genotypes were divided into two clusters that formed landraces/ecotypes, respectively breeding lines and varieties. This indicated a change in genetic structure due to breeding. The most comprehensive marker system used for the assessment of genetic diversity are microsatellites (SSRs, simple sequence repeats), as they have many desirable properties, such as abundant genome distribution, the ability to form multiplexes, and codominance, i.e., the ability to distinguish between homozygotes and heterozygotes [26]. Molecular approaches improve germplasm conservation efforts and increase the use of plant genetic resources [27]. Furthermore, combining morphological and molecular markers is an excellent approach for selecting optimal genotypes for breeding, while maintaining adequate genetic diversity [28], and knowledge of genetic diversity is the basis for the effective management and utilisation of plant germplasms.

This study aimed to investigate the genetic structure of the yellow mustard core collection and to evaluate the genetic diversity among the yellow mustard accessions. A thorough investigation of the molecular properties of this diversity yields a group of the most diverse genotypes for use in breeding programmes. Marker-assisted selection using polymorphic markers among parents helps with the finding and choosing of suitable parents for hybridisation and backcross breeding for future use.

## 2. Results and Discussion

### 2.1. Microsatellite Polymorphism and Genetic Diversity of Genetic Resources

The wide collection of genetic resources for yellow mustard (*S. alba*) was represented by 198 unique accessions (registered varieties, landraces, and breeding materials). Microsatellites generated 113 alleles and, of the total alleles, 44 were polymorphic and detected in 15 loci, with an average of 2.93 alleles per locus and a maximum of six alleles in D3_A, P7, and P35_A loci. Five loci (BoREM1b_B, BoREM1b_C, BolAB19TF, P30_C, and P35_B) were classified as uninformative loci with MAF < 0.01. Only one allele was detected in these loci of one to three individuals/accessions. BoREM1b_B was successfully amplified in genotype 9 (Morocka, landrace) and BoREM1b_C in genotypes 51 (SIN 10/79, breeding material), 167, and 177 (Asta and Raduga, new registered varieties). BolAB19TF was amplified in genotype 130 (Sabon, new registered variety), P30_C in genotypes 169 and 186 (Rumba and Olga, new registered variety), and P35_B in genotype 174 (Bamberka, new registered variety). This study is part of a project focused on yellow and coloured mustard, and only the markers successfully amplified in yellow mustard are presented in Table 1. Samples with the occurrence of uninformative loci were also included in these results, because these genotypes with the presence of even this one allele differ from the set of other varieties. These unique/less common alleles are typical for a particular variety and lead to higher values of genetic diversity (see Appendix A and WAM value). It is also noticeable that these less frequent alleles, with the exception of the Marocka landrace, occur in newly bred varieties and breeding materials. The size of the amplified alleles was in the range of 125–314 bp. Eleven loci were medium to highly polymorphic, with PIC values varying from 0.19 to 0.65. The highest PIC values were observed in loci D3_A (0.65) and P35_A (0.52), and also in loci BoPC34 (0.51) and P381 (0.50), characterised by a low number of alleles per locus. The level of observed heterozygosity was in the range of 0.12 to 0.83, with an average of 0.21. The highest observed heterozygosity was also recorded in loci with a high polymorphic index, namely D3_A (0.83), BoP34 (0.73), and P35_A (0.53). Nei’s gene diversity index ranged between 0.11 and 0.68, with an average of 0.25. Such a high level of polymorphism, compared to other oil seed Brassicas, may be explained by a higher level of cross-pollination in yellow mustard [29]. Detailed results of the microsatellite analysis are given in Table 1.

Allele sizes calculated from CE electropherograms were identical for the internal controls and stable in all repetitions. This uniformity and reproducibility for all microsatellites were indicators of correctly set protocols of molecular analyses. The microsatellites used in this study were originally designed for *Brassica oleracea* [30], *B. rapa* ssp. *chinensis*, *B. juncea*, and *B. nigra* genetic resources [31]. Our results showed very good cross-transferability and possible utilisation of these markers in yellow mustard. Successful amplification of specific fragments from the more distantly related Brassica species, including *S. alba* using primers designed for *B. napus*, were also reported [19]. Still, they did not analyse the diversity of microsatellites in yellow mustard genetic resources in more detail. Therefore, this study presents the first results of the analysis of microsatellites in a large set of genetic sources of yellow mustard and shows relatively high variability in the genetic resources, as well as the cross-transferability of markers and the occurrence of unique alleles.

The results of the AMOVA (Analysis of molecular variance) showed that the highest proportion of diversity was found within the analysed set of genetic resources (87.13%) and that the proportion of diversity and differentiation between types of genetic resources (4.6%) or between genotypes within types of genetic resources (8.27%), although significant, was always much lower (Table 2). Likewise, the null hypotheses were rejected (for parameters Φ_CT_, Φ_SC_—*p* < 0.001, and Φ_ST_—*p* < 0.005). Thus, there are genetic differences between types of genetic resources, as well as between individual accessions of yellow mustard within the analysed set of genetic resources.

The genetic diversity of the yellow mustard germplasms was calculated using DARwin 6 software. The dendrogram in Figure 1a shows the results of the weighted neighbor joining analysis, and, in Figure 1b, the results of the principal coordinates analysis. The extent of genetic variation in the analysed collection was in the range of 0.00–1.00 (Appendix A). The furthest distances were between accessions ‘47-635-YE 13′ and ‘Braco’. Additionally, the seven other most different/distant pairs of accessions always included the Chinese ‘47-635-YE 13′ genotype. Both the genetic diversity and dissimilarity range were much higher than the values reported by Fu et al. [17], given the more considerable extent of the analysed collection and the marker system used. The high level of dissimilarity between the accessions analysed was documented by the WAM value, which was 33.82%. A higher WAM value corresponded with the pattern of genotype distribution after factorial (PCoA) analysis (Figure 1b). In our previous study, focused on *Brassica napus* [32], we observed WAM values that were significantly lower (22.50% for SSR markers). These results showed the importance of extensive collections of genetic resources for preserving genetic diversity. The results of other studies conducted on mustard showed significantly lower values of genetic diversity [33], probably due to the low diversity of the collection and the small number of samples, which was incomparably lower than in this study.

Based on microsatellite analysis, it was possible to identify a large proportion of the analysed accessions unambiguously, but even when a relatively high number of markers were evaluated, several accessions of different origins had precisely the same patterns of microsatellite markers. There were two main groups of accessions with the same pattern of microsatellites. The first group included genotypes from Germany, Denmark, Netherlands, and Poland, but also genotypes from Australia and an old landrace from the Czech Republic/former Czechoslovakia (‘ABA’, ‘Borowska’, ‘BRSCHW 55705′, ‘Česká krajová’, ‘Perovska’, ‘Sito’, and ‘King’). The second block of genotypes with the same pattern of microsatellites mainly included Czech varieties (‘Andromeda’ and ‘Severka’), as well as varieties from Germany, Austria, and the Netherlands (‘Ascot’, ‘Emergo’, and ‘Mirly’). There were also other groups or pairs of indistinguishable accessions, such as ‘OP-2408′ and ‘OP-2415′, or ‘Olga’ and ‘OP-2291′, originating in one breeding station, but also cases have been reported where landraces have the same microsatellite profile as modern varieties, e.g., ‘Přerovská bílá’ vs. ‘Ludique’ and ‘Budakalászi sárga’, or ‘Dr. Schneider’ vs. ‘Concerta’ and ‘Esprit’. A relatively narrow genetic base should explain these results, and unrelated genotypes with an identical pattern of microsatellites are used in different breeding programmes. These findings also reveal the need for better management of breeding populations based on genetic analysis, and further improvement is also increasingly dependent on the evaluation of genetic diversity resources and breeding populations. Breeding materials from the TILLING population showed significantly higher variability and distinct differences from other genetic resources. These materials were created precisely with the goal of extending genetic variability and their further utilization in breeding programmes (accessions 188–198 in the lower left part of the dendrogram in Figure 1a). Fu et al. [17] reported a limited range of genetic resources used in mustard breeding, and the importance of germplasm protection and narrowing genetic variation has been discussed for other cruciferous oilseeds, including mustards [8,33,34]. For this reason, it is appropriate to use molecular markers in selection programmes. Genetic resources (paternal components used for crossing) should be selected not only according to their phenotypic values but also in terms of genetic diversity or distance, and microsatellites can be used as effective practical tools to provide molecular data and evaluate the genetic relationship [35,36,37].

### 2.2. Genetic Structure of Yellow Mustard Varieties

Bayesian clustering algorithms were performed in the STRUCTURE 2.3.4 software to describe the genetic structure better and to explain the results from previous statistical analyses. Although this approach has been highly frequented in ecological studies [38] in the last period, it is also used for agricultural crops including oil seed brassicas, such as *B. napus* [32], *B. juncea* [23], and *B. carinata* [25]. Through the STRUCTURE software, we determined the likelihood of the data for several different K, with K ranging from one to seven, to eventually see the plateau pattern in the diagnostic plot. During these computations, we set the burnin period to 100,000, and 150,000 MCMC repeats after burnin. Such a setting seemed to be sufficient, as we attained good mixing and a stable process in equilibrium, with no excessive variations in the ln(alpha) parameter estimates. The parameter alpha (the quantification of relative levels of admixture between populations) was allowed to vary for each population to obtain a more accurate estimate of the ancestry. The starting value for the Dirichlet parameter alpha was set to one with a uniform prior. We applied the model with admixture and correlated allele frequencies to attain greater power, in order to detect potential distinct populations that could be closely related. The distributional parameter lambda was computed in the initial pre-run (with lambda set to one), and the inferred value was used in sequential runs (the quantification of independence between markers in terms of allelic frequency distribution). Other hyperparameters were left at their default value. We did not use the LOCPRIOR parameter in our simulations. Consequently, for optimal K = two, we estimated the individual Q-matrix [38,39,40]. The detailed outputs of cluster number estimations are given in Appendix A and Figure 2. The results and outputs of the STRUCTURE software (Q1/Q2 values) are given in Figure 3 and Appendix A.

The accessions were classified into particular groups, according to Q1/Q2 values (Figure 3a). The mean alpha value was estimated at 0.0602, indicating substantial mating barriers between breeding populations (varieties) and reproductive isolation, due to the limited exchange of genetic resources between individual breeders. The mean values of Fst_1 and Fst_2 were 0.3482 and 0.1916, respectively. Also, these values indicated considerable genetic differentiation among the accessions, as was reported previously [41].

The detailed analysis of the raw data and outputs from the STRUCTURE software indicated the presence of a transition zone between the two main clusters estimated, according to parameter Delta K = two. Therefore, detailed classification of accessions according to Q1/Q2 values (Figure 3b) led to the formation of three blocks: a/the first block included accessions with Q1 values in the range of 0.008–0.095; b/the second block formed a transition zone with Q1 values in the range of 0.107–0.396; c/the third block included accessions with Q1 values in the range of 0.515–0.991. Most accessions (113; 57.1%) were assigned to the first block; the third block comprised 73 accessions (36.8%) and 12 accessions (6.1%) formed the transition group. In the first most diverse groups, there were both earlier varieties and landraces, as well as modern varieties and breeding materials. In the second and third groups were landraces and bred varieties, but not new breeding materials. These results from the analysis of the genetic structure were supported by the outputs from the AMOVA, where, although genetic differentiation between types of genetic resources and between individual accessions of yellow mustard within the collection of genetic resources was statistically significant, the proportion of variance between types of genetic resources (landrace, variety, and breeding material) was very low (4.6%). On the contrary, the proportion of variance between genotypes within the analysed set of genetic resources was the highest (87.13%). A more detailed analysis of genetic structure results shows somewhat more interesting outputs than in the case of the N J analyses. It was evident that varieties originating from some countries or breeding programmes have similar or the same genetic structures. However, it is more common that, according to the genetic structure, the varieties from one country of origin are classified into two different clusters, and the year of origin of the variety is significantly reflected, because the main factor that distinguishes these clusters is the parameter “year of origin/registration”. It also often reflects changes in the breeding programme, the use of different genetic resources, or the focus on the breeding of new traits (Figure 4).

This distribution of varieties is evident, for example, in Hungarian varieties, where the most recent and older varieties differ significantly in genetic structure. A similar characteristic can be seen in varieties from the Netherlands, Belgium, Denmark, France, Sweden, and Romania. Different characters and explanations may be given for grouping German and Czech varieties. Four German varieties (‘Seco’, ‘Semper’, ‘Setoria’, and ‘Sigri’) belonged to the same group with closely similar Q1 values. They were bred at one breeding station, and their genetic structure reflects the significant similarity of the genetic resources used in breeding programmes. In the case of Czech varieties, both landraces (‘Přerovská bíla’, ‘Česká krajová’), the older variety ‘Zlata’, and modern varieties from the breeding stations Opava and Krukanice belong to the same group. The second cluster comprises current modern varieties from breeding programmes in Selton and BOR (‘Agent’ and ‘Veronika’). In these cases, there is a strong influence of a limited set of genetic resources, and molecular marker methods and the evaluation of genetic diversity can help to appropriate the management of genetic resources and to spread genetic variability and shifts in genetic structures. In this study, some landraces and older varieties were not significantly different from modern varieties. However, it is still evident that the origin of the variety, country of origin, or the breeding programme significantly influences the genetic structure. Similar results were presented in oil seed brassicas [32] or in *B. juncea* [42] and *B. carinata* [34] genetic resources. In contrast, in *B. rapa*. no significant loss of genetic diversity was recorded, although most total variation was attributed to within-cultivar variation in cross-pollinated cultivars [43]. The large geographical variation of genetic resources is a significant source of genetic variation, and high variation among subpopulations of genetic sources has been observed in *B. juncea* [44].

An essential problem in current breeding is the insufficient range of genetic diversity and the related limited availability of interesting genetic resources, such as donors of required traits and characteristics [45]. The depletion and narrowing of genetic diversity occurred in the context of the development of modern breeding in the early 20th century, due to intensive breeding focused on a limited range of traits and the influence of the cultivation of high-yielding and high-quality, but often relatively closely related, varieties of agricultural crops [46,47]. One possibility to expand the range of genetic variability is TILLING technology [48]. This modern approach, combining classical mutagenesis with NGS techniques and the detection of genes carrying mutations, seems to be a very promising breeding method. Plants carrying a change/mutation in the gene of interest are potential donors of a new characteristic and are incorporated into breeding programmes [49]. The results of the microsatellite analysis show, in this study, a significant spread of genetic variability in the mutant population. All plants from the mutant population were incorporated into the yellow mustard breeding programme and, in the following years, qualitative traits and their possible changes due to mutation will be assessed.

## 3. Materials and Methods

**Plant material and DNA extraction.** For the evaluation of genetic diversity, a pilot set of 198 high-value accessions was chosen from the collection of yellow mustard kept at the OSEVA Development and Research in Opava, Czech Republic. These accessions were selected based on their value for current breeding programmes, including modern and old registered varieties, landraces, breeding material, and accessions from the TILLING population. The accessions came from various countries, representing a wide range of spring-type cultivars with a wide range of seed colours. The list of genetic resources, their accession numbers in the gene bank, and their country of origin are given in Appendix A. The seeds were seeded on a horticultural peat substrate and grown under laboratory conditions at a temperature of 21 °C and 16/8 h for five days. The bulk of 36 cotyledons was sampled and gently dried in silica gel. The dried material was homogenised using glass beads in a Retsch^®^ Mixer Mill MM400 (Haan, Germany) for 30 s at a maximal frequency (30 Hz). The DNA was extracted by a modified CTAB PVP method [50], which is more time-consuming, but the obtained DNA is very pure and highly concentrated.

**Microsatellite amplification.** A total of 15 microsatellite markers were used: BoREM1b_A, BoREM1b_B, BoREM1b_C, BolAB19TF, BoPC34 [51], P381, D3_A, D3_B, P7, P9_B, P9_C, P30_B, P30_C, P35_A, and P35_B [31]. Three primers were used in each PCR reaction, one universal FAM-labelled primer (Ba02 6-FAM or Ba03 6-FAM), a forward primer extended by the sequence of the labelled primer, and a reverse primer [52]. The PCR reaction was carried out in a 10 µL, in 1× reaction buffer (75 mM Tris—HCl, pH = 8.8, 20 mM (NH_4_)_2_ SO_4_, 0.01% Tween 20, 2.5 mM MgCl_2_, 200 µM dNTPs), 10 pmol of each primer, 0.5 U of Taq Purple DNA polymerase PPP Master Mix (Top-Bio, Vestec, Czech Republic) or 2× Master Mix with standard buffer (NEB, Ipswich, MA, USA), 2× BSA, and 50 ng of template DNA. The amplification was carried out on a Biometra TProfessional thermocycler (Göttingen, Germany) in the following temperature profile: initial denaturation of 5 min at 94 °C, 31 cycles of 60 s at 95 °C, 60 s at 52–59 °C (depending on the primer used), and 60 s at 72 °C/68 °C, with a final elongation of 15 min at 72 °C/68 °C.

**Data Analysis.** The results of the PCR analysis were evaluated by fragment analysis on an ABI 3500 genetic analyser (Applied Biosystems, Waltham, MA, USA). Molecular data were analysed using GeneMapper version 3.0 software. The weighted arithmetic mean of genetic distances (WAM) was calculated based on frequency values in the appropriate class of genetic diversity [32]. Diversity measures were calculated in adegenet for the R software version 4.2.2 [53,54]. They consisted of the number of alleles (NA), major allele frequencies (MAF), observed heterozygosity (Ho), expected heterozygosity (He), and the polymorphism information content (PIC). To determine genetic diversity among and within genetic resources, AMOVAs [55] were performed with the package “pegas” in R [56]. The null hypotheses associated with differentiation parameters (Φ) were as follows: there is no genetic difference between types of genetic resources (Φ_CT_ = 0), between genotypes within types of genetic resources (Φ_SC_ = 0), or between genotypes within collection of genetic resources (Φ_ST_ = 0) [55]. The data were statistically evaluated using traditional genetic diversity metrics in MVSP 3 (Kovach Computing Services, Pentraeth, UK) and DARwin 6 (CIRAD, F) software. Intending to estimate the population/cluster number, we used the Bayesian clustering approach, involving MCMC estimation, and followed Evanno et al. [40] and Kolář et al. [38]. STRUCTURE software was used to assess the genetic structure of analysed germplasms [39].

## 4. Conclusions

The issue of genetic diversity, its conservation, protection, and exploitation are the key points in plant breeding. In this study, we focused on the detailed description and molecular analysis of the genetic variability of a large collection of yellow mustard genotypes. High values of observed heterozygosity, diversity index, and WAM index indicated a high level of polymorphism between individual accessions. Also, outputs from AMOVA supported these findings and showed a low proportion of variance (4.6%) between types of genetic resources (landrace, variety, and breeding material) and a high proportion of variance between genotypes within the analysed set of genetic resources (87.13%). In addition to the standard descriptive characteristics, two other procedures were applied, for an overall assessment of genetic variability and the degree of diversity of the genetic resources: PCoA analysis and Bayesian clustering algorithms. The results show a more valuable interpretation and highlight another way of processing the genetic diversity data and indicated mating barriers between varieties and reproductive isolation, due to the limited exchange of genetic resources between breeders. The range of genetic diversity was much higher than reported in other studies, concerning the large and variable collection of genetic resources. These results showed the importance of the proper management of genetic resources for the preservation of genetic diversity. Analysis of genetic structure may lead (as shown in our example) to a clarification of the grouping of accessions (genetic resources and varieties) and recommendations for expanding the genetic background in breeding programmes. The choice of parental components based on phenotypic characteristics and genetic diversity facilitates the selection of genetic resources for breeding programmes and can contribute to the success of breeding.

## Figures and Tables

**Figure 1 plants-12-04026-f001:**
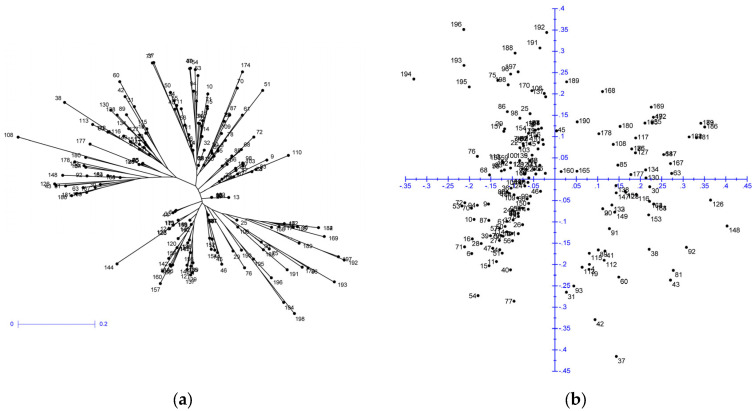
Outputs of statistical analyses of SSR markers in 198 *S. alba* accessions: (**a**) results of cluster analysis (weighted N J analysis); (**b**) results of principal coordinates analysis (PCoA).

**Figure 2 plants-12-04026-f002:**
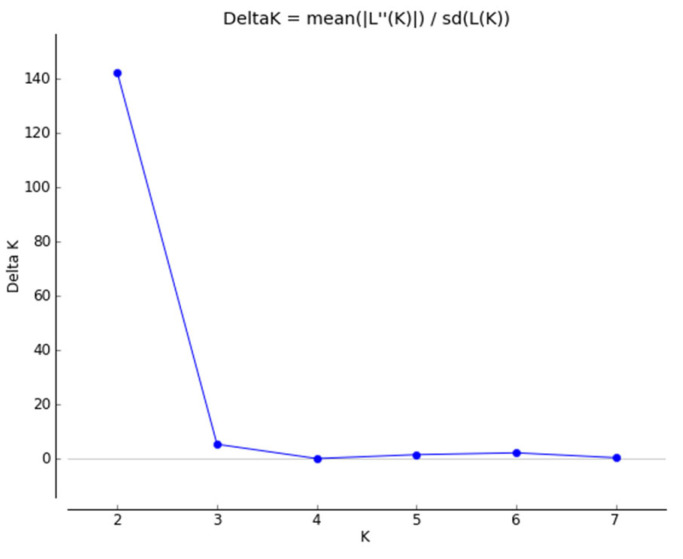
The magnitude of Delta K as a function of K.

**Figure 3 plants-12-04026-f003:**
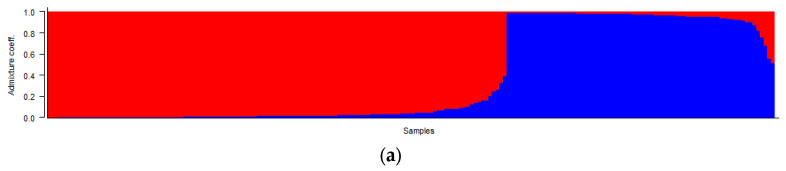
Detailed classification of accessions according to Q1 (blue)/Q2 (red) values: overall overview of the distribution of Q1/Q2 values in the analysed set of genetic resources (**a**); detailed analysis of genetic structure with marking of individual accessions (**b**).

**Figure 4 plants-12-04026-f004:**
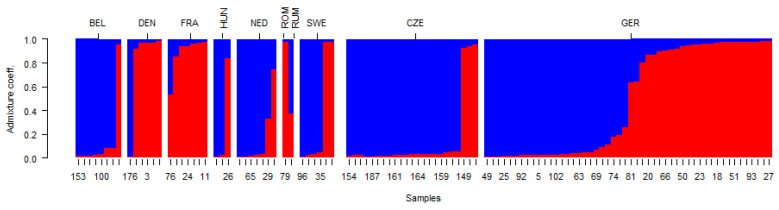
Classification of accessions from selected countries of origin, according to Q1 (red)/Q2 (blue) values and the year of registration.

**Table 1 plants-12-04026-t001:** Detailed results of microsatellite analyses.

Microsatellite Marker	Number of Individuals	PIC	Na (Number of Alleles)	Number of Alleles in Population	Length of Alleles in Bp	Median and Max. Frequency of Alleles	Ho (Observed Heterozygosity)	He (Nei, 1973; Expected Heterozygosity)	Fis (Inbreeding Coefficient)	1-D (Simpson Index)	Hexp (Nei. 1978; Gene Diversity)	Evenness
BoREM1b _A	198	0.19	2	209	173–175	104.5; 187	0.12	0.11	−0.06	0.11	0.11	0.50
BoREM1b _B	198	0.00	1	3	181	3; 3	0	0	-	-	-	
BoREM1b _C	198	0.00	1	3	187	3; 3	0	0	-	-	-	
BolAB19TF	198	0.00	1	1	314	1; 1	0	0	-	-	-	
BoPC34	198	0.51	3	323	149–151	148; 172	0.73	0.50	−0.45	0.50	0.50	0.94
P381	198	0.50	2	21	213–229	105; 11	0.05	0.50	0.90	0.50	0.51	1.00
D3 _A	198	0.65	6	342	155–163	33.5; 149	0.83	0.67	−0.24	0.67	0.67	0.85
D3 _B	198	0.44	5	101	167–171	10; 73	0.20	0.39	0.50	0.38	0.68	0.54
P7	198	0.27	6	119	141–156	3.5; 101	0.11	0.21	0.49	0.21	0.21	0.39
P9 _B	198	0.38	3	184	125–138	32; 141	0.28	0.29	0.05	0.27	0.27	0.54
P9 _C	198	0.35	4	231	145–160	23; 184	0.25	0.23	−0.06	0.23	0.24	0.49
P30 _B	198	0.32	2	5	166–174	2.5; 4	0	0.32	1.00	0.32	0.36	0.72
P30 _C	198	0.00	1	2	192	2; 2	0	0	-	-	-	
P35 _A	198	0.52	6	109	149–161	3.5; 68	0.53	0.47	−0.12	0.46	0.47	0.63
P35 _B	198	0.00	1	1	173	1; 1	0	0	-	-	-	

**Table 2 plants-12-04026-t002:** Analysis of molecular variance (AMOVA) results for yellow mustard genetic resources (between and within types of genetic resource) with df, sums of squares (SS), mean squares (MS), genetic variance estimates (Sigma), proportion of variance (%) attributed to the different levels in the spatial hierarchy, Phi (Φ) statistics that are analogous to Wright’s F-statistic, and *p* values (9999 permutations).

	df	SS	MS	Genetic Variance Estimates (Sigma)	Proportion of Variance (%)	Phi (Φ)	*p*- Values
Between types of genetic resources	2	19.41	9.705	0.143	4.602	Φ_CT_ = 0.0460	<0.0001
Between genotypes within types of genetic resources	31	123.07	3.970	0.257	8.272	Φ_SC_ = 0.0867	<0.0001
Within genetic resources	164	414.12	2.707	2.707	87.126	Φ_ST_ = 0.1287	0.0028
Total	197	556.60	2.992	3.107	100		

Type of genetic resource = landrace/variety/breeding material.

## Data Availability

Additional data files and project outputs are available at http://biocentrum.zf.jcu.cz/laborator.php?sub=metodiky (accessed on 20 November 2023).

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
