# Peer review of "Microsatellite Markers: A Tool to Assess the Genetic Diversity of Yellow Mustard (Sinapis alba L.)"

_plants, 2023, doi:10.3390/plants12234026_

Round 1

Reviewer 1 Report

Comments and Suggestions for Authors

Reviewing opinions for the manuscript titled by Microsatellite markers: new era tool to assess the genetic diversity of yellow mustard (Sinapis alba L.)

In the present study, genetic diversity and genetic structure of a collection of 187 yellow mustard accessions representing registered varieties, landraces, and breeding materials, was characterized using SSR markers. The results revealed that a high level of polymorphism among the tested materials. The findings will provide value references to the yellow mustard breeding in the future.

The following opinions were put forward to when the authors revise their manuscript.

1)      Abstract: should be improved, including your main findings, clustering, PCoA, structure.

2)      Table 2, change to be the supplementary files.

3)      Part 2.3 Pattern of genetic variability in mutant population: It is primary result. I suggest the author consider to delete this part in the manuscript. After more research works has been done, you could publish this part in the future. In addition, the mutation dependent on the detection tool, for example, if you use SNP by GBS, you will detect more mutation. 

4)      Conclusions: The conclusion would be improved, more concise. not included references here.

5)      References: There are too much references cited, it can be reduced.

Comments on the Quality of English Language

Author Response

  • Abstract: should be improved, including your main findings, clustering, PCoA, structure.

The abstract was rewritten, our main findings, results of population diversity and genetic structure were added according to comments

  • Table 2, change to be the supplementary files.

Table 2 was added to supplementary files

  • Part 2.3 Pattern of genetic variability in mutant population: It is primary result. I suggest the author consider to delete this part in the manuscript. After more research works has been done, you could publish this part in the future. In addition, the mutation dependent on the detection tool, for example, if you use SNP by GBS, you will detect more mutation. 

Part 2.3 was removed and the results will be prepared into the formo of separate manuscript. Several key genotypes were evaluated in this study to demonstrate widening of genetic diversity in TILLING population of S. alba.

  • Conclusions: The conclusion would be improved, more concise. not included references here.

Part of the conclusion has been shortened and all references have been removed.

  • References: There are too much references cited, it can be reduced.

References have been reduced according to reviewer recommendations and only key references are included in manuscript.

Reviewer 2 Report

Comments and Suggestions for Authors  

Title Microsatellite markers: new era tool to assess the genetic diversity of yellow mustard (Sinapis alba L.)

Authors Eva Jozová et al

In this study, Microsatellite markers were used for genetic diversity evaluation and assessment of genetic structure in a germplasm collection of yellow mustard. The manuscript is well-structured, clear, and presents important results. However, I have some comments to improve the overall quality of the manuscript.

Title needs to be modified. The use of Microsatellite markers technology to study genetic diversity is already very extensive. It is no longer appropriate to call it a new era tool.

Lines 87-92: The introduction is brief, a little short compared to the abundant literature on the subject.  The research progress on polymorphism in mustard plants needs to be described in detail, including what results have been achieved.

Line 52: Verify the insertion position of reference 12.

Line 107: Sinapis alba L. has already been mentioned before, so it should be abbreviated.

Line 108: "187 unique accessions (registered varieties, landraces, and breeding materials)".

Table S1  should include  to demonstrate the sources of germplasm resources.

The clarity of Figure 5 is not sufficient, and the numbers in the figure are not discernible. Additionally, there is some background color that can be removed.

Lines 358-360: The selection of microsatellite markers is based on literature. It would be helpful to know why the authors did not design their own markers or the reasons for selecting these markers.

The Conclusion should be summarized briefly. Conclusion generally do not need to include references.

Author Response

  • Title needs to be modified. The use of Microsatellite markers technology to study genetic diversity is already very extensive. It is no longer appropriate to call it a new era tool.

The title has been modified.

  • Lines 87-92: The introduction is brief, a little short compared to the abundant literature on the subject.  The research progress on polymorphism in mustard plants needs to be described in detail, including what results have been achieved.

References have been reduced according to reviewer recommendations and only key references are included in manuscript. Results of analyses of population and genetic structure in genetic resources of coloured mustards were described in details.

  • Line 52: Verify the insertion position of reference 12.

Position of references 12 was corrected and was inserted in order of proper reference.  

  • Line 107: Sinapis alba L. has already been mentioned before, so it should be abbreviated.

The presentation of the species name has been modified throughout the text. 

  • Line 108: "187 unique accessions (registered varieties, landraces, and breeding materials)".

Table S1  should include  to demonstrate the sources of germplasm resources.

Table S1 was supplemented with descriptive data. 

  • The clarity of Figure 5 is not sufficient, and the numbers in the figure are not discernible. Additionally, there is some background color that can be removed.

Figure 5 as well as the whole chapter 2.3 was removed – according editor suggestions

  • Lines 358-360: The selection of microsatellite markers is based on literature. It would be helpful to know why the authors did not design their own markers or the reasons for selecting these markers.

The markers used were approved as official marker systém for the evaluation of the genetic resources of Brassica oilseeds and vegetables deposited in the gene bank in the Czech Republic.

Markers published in previous publications were also used for the reason that comparison with previous results was possible. Microsatellite markers were selected based on the characteristics reported in the cited publications. PIC characteristics, number of alleles and number of polymorphic alleles were considered most important. If the authors had designed their own microsatellite markers, it would not have been possible to compare the results with previously published results.

  • The Conclusion should be summarized briefly. Conclusion generally do not need to include references.

Part of the conclusion has been shortened and all references have been removed.

Reviewer 3 Report

Comments and Suggestions for Authors

Reviewer’s Comment / Report

The manuscript # plants-2698752 entitled “Microsatellite markers: new era tool to assess the genetic diversity of yellow mustard (Sinapis alba L.)" has been reviewed.

The authors have done significant work on genetic diversity evaluation and assessment of genetic structure in a germplasm collection of yellow mustard, Sinapis alba L. However, the study failed to conclude the study as the author mentioned that they have used collection of genetic resources represented 187 registered varieties, landraces, and breeding materials. The study does not provide sufficient information for the samples which belong to varieties, landraces, and breeding materials.

Similarly, the study mentioned that some landraces and older varieties were not significantly different from modern varieties and evident that the origin of the variety, country of origin, or the breeding program significantly influences the genetic structure. However, it is difficult to understand their variation based on STRUCTURE or population cluster.

Comments on the Quality of English Language

Can be improved 

Author Response

The authors have done significant work on genetic diversity evaluation and assessment of genetic structure in a germplasm collection of yellow mustard, Sinapis alba L. However, the study failed to conclude the study as the author mentioned that they have used collection of genetic resources represented 187 registered varieties, landraces, and breeding materials. The study does not provide sufficient information for the samples which belong to varieties, landraces, and breeding materials.

Similarly, the study mentioned that some landraces and older varieties were not significantly different from modern varieties and evident that the origin of the variety, country of origin, or the breeding program significantly influences the genetic structure. However, it is difficult to understand their variation based on STRUCTURE or population cluster.

Conclusions were rewritten

Table S1 was supplemented with descriptive data and type of genetic resource was added.

We also modified Figure 3, it was supplemented with a detailed results of the analysis of the genetic structure, and the IDs of individual accessions of genetic resources are also added. These results are described in the text, where the most significant examples are given - the same character of landraces, modern varieties or influence of origin/breeding firm – e.g. from Fig 3B can be seen that landraces (2-Ceska krajova, 12-Prerovska bila), old variety (154-Zlata), modern/recent varieties (152-Polarka, 157-Otava) and Breeding material (164-OP2415) are gouped together, they are in the same cluster with very similar Q1/Q2 values. On the other side modern varieties 111-Agent and 150-Veronika are located in second cluster. The reason of this similarities of landraces, old and modern varieties is low genetic diversity between genetic resources used in particular breeding programme. So proper management of genetic resources can help breeders to widening genetic variability and breeding of new varieties.

Round 2

Reviewer 1 Report

Comments and Suggestions for Authors

The authors have revised their manuscript according to my suggestion. I recommend to accept this ms.

Comments on the Quality of English Language

The manscript has been improved a lot. Its English is good.

Author Response

The reviewer had no comments, we answered all questions in the previous revision of the manuscript.

Reviewer 3 Report

Comments and Suggestions for Authors

The manuscript's conclusion posits a notably broader range of genetic diversity within the studied collection of genetic resources compared to previous studies. However, the correlation between this conclusion and the content provided is somewhat limited. Table 1 stands as the primary reference supporting this conclusion, demonstrating instances of extremely low PIC values (0) in certain loci, suggestive of minimal allelic variation. This finding raises questions about the adequacy of the analysis in capturing the full spectrum of genetic diversity.

To substantiate the claims made regarding the extensive genetic diversity, further analyses, such as AMOVA (Analysis of Molecular Variance) and other relevant methodologies, would significantly enhance the manuscript. These additional analyses could provide a more comprehensive understanding of the allelic distribution and aid in corroborating the asserted range of genetic diversity. Strengthening the study with such analyses would better align the conclusions with the empirical evidence presented

Author Response

The manuscript's conclusion posits a notably broader range of genetic diversity within the studied collection of genetic resources compared to previous studies. However, the correlation between this conclusion and the content provided is somewhat limited. Table 1 stands as the primary reference supporting this conclusion, demonstrating instances of extremely low PIC values (0) in certain loci, suggestive of minimal allelic variation. This finding raises questions about the adequacy of the analysis in capturing the full spectrum of genetic diversity.

the explanation is given in the new added text on lines 121-133

low PIC values resulted from presence of rare alleles in some genotypes (mostly new varieties and breeding materials), these alleles were unique for the particular genotypes and even if they occur in non-informative loci they are important in terms of variety identification and lead to higher calculated values of genetic diversity

To substantiate the claims made regarding the extensive genetic diversity, further analyses, such as AMOVA (Analysis of Molecular Variance) and other relevant methodologies, would significantly enhance the manuscript. These additional analyses could provide a more comprehensive understanding of the allelic distribution and aid in corroborating the asserted range of genetic diversity. Strengthening the study with such analyses would better align the conclusions with the empirical evidence presented.

results from AMOVA are included in text – new Table 2, explanation text in results – chapter 2.1 as well as chapter 2.2

results from AMOVA supported outputs from STRUCTURE and we can better explain the grouping of genotypes/accessions – we found no statistically significant differentiation between types of genetic resources, so variation between types of genetic resources is smaller than within genetic resources

Round 3

Reviewer 3 Report

Comments and Suggestions for Authors

It sounds like the author made substantial improvements to the manuscript based on the suggestions provided by the reviewer. If the revisions addressed the concerns and suggestions raised by the reviewer and the manuscript now meets the required standards, it's possible that it could be accepted in its present form.